# Burden of Care of Family Caregivers for People Diagnosed with Serious Mental Disorders in a Rural Health District in Kwa-Zulu-Natal, South Africa

**DOI:** 10.3390/healthcare11192686

**Published:** 2023-10-06

**Authors:** Jabulile Thembelihle Ndlovu, Kebogile Elizabeth Mokwena

**Affiliations:** 1Department of Public Health, Sefako Makgatho Health Sciences University, Pretoria, Ga-Rankuwa 0204, South Africa; kebogile.mokwena@smu.ac.za; 2Substance Abuse and Population Mental Health, Sefako Makgatho Health Sciences University, Pretoria, Ga-Rankuwa 0204, South Africa

**Keywords:** Zarit Burden Interview scale, burden of care, home care-giving, severe mental illness, rural setting

## Abstract

An estimated 6% of the world population has serious mental illness, with one in four families having a member with some form of psychiatric disorder, who is mostly cared for by their relatives within a family setting. Although care-giving in a home setting is reported to be associated with significant mental distress, the burden of such distress is rarely measured. The purpose of this study was to quantify the burden of care among family caregivers of relatives with serious mental disorders, as well as to explore possible associations between the caregiver burden of care and a range of caregiver and Mental Health Care User (MHCU) variables in a rural district in Kwa-Zulu Natal, South Africa. The Zarit Burden Interview (ZBI) scale was used to collect data from 357 caregivers, and STATA 14 was used to analyze data. The ages of the sample ranged from 18 to 65 years, with a mean of 50.29, and the majority (86%) were female and unemployed (83%). The ZBI scores ranged from 8 to 85, with a mean of 41.59. The majority (91%) were found to be affected by family caregiver burden, which ranged from mild to severe. Using the Pearson Chi-square test of association (*p* = 0.05), variables that were significantly associated with the burden of care were clinically related (caregiver self-reported depression, MHCU diagnosis, recent relapse of the MHCU), socio-economic (caregiver family monthly income, MHCU disability grant status and MHCU employment status) and socio-demographic (MHCU gender and MHCU level of education). The prevalence of the burden of care is high and severe, and the scarcity of resources in families and communities contributes to the high burden of care in these rural communities.

## 1. Introduction

Mental disorders contribute to an estimated 14% of the global burden of diseases, with the highest burden in developing countries [1]. The burden continues to negatively impact the economic profile of affected countries, with resultant declines in productivity at both the national and individual levels, which is why they need national attention. While psychiatric conditions are responsible for little more than 1% of deaths globally, they account for almost 11% of the burden of disease [2]. Evidence of the negative impact of mental disorders on the health and wellbeing of the family caregivers continues to emerge, which is often reported to be worse in cases of depressive disorders or embarrassing behavior [3,4].

Among many African societies, caring for a family member who needs constant support, such as one who has chronic mental disorder, has been traditionally shared with other members, including extended families [5], which has been beneficial for both the caregivers and the person with chronic mental illness. However, changes in the social structures continue to shrink the extended family as it evolves towards smaller nuclear families [6]. The relatively small nuclear families are left without financial and/or social support, thus becoming more vulnerable to an unmanageable burden of care. This results in many caregivers bearing their physical, emotional, spiritual, and financial burdens in solitude [7], as they miss out on the traditional family support networks. 

The COVID-19 pandemic has worsened the experiences of both the patients with mental illness and their caregivers. The pandemic has not only increased the global burden of communicable diseases, but has also presented long-term economic and social consequences that have increased the prevalence of both depression and anxiety disorders. The COVID-19 impact indicators and shifting priorities of governments worldwide have substantially impacted the mental health status of the world population, including the ability to care for family members at the household level. In particular, decreased social interactions, lockdowns, stay-at-home orders, decreased public transport, school and business closures, as well as subsequent loss of livelihood, loss of jobs and decreases in economic activity, have all negatively affected the mental health of the world population. The pandemic has thus created an environment in which many determinants of mental health play out, which includes caring for the mentally ill by their relatives at home, which remains a great concern [8].

South African legislation and policies that are intended to reduce psychosocial disability and promote mental health include the Mental Health Care Act no. 17 of 2002 (MHCA), the International Covenant on Economic, Social and Cultural Rights (ICESCR), as well as the African Charter on Human and People’s Rights on the Rights of Persons with Disability in Africa.

In South Africa, a significant portion of the budget for mental health services is traditionally used for in-patient care, leaving community or family-based care structures unfunded [9]. Therefore, the majority of people with mental illness who do not need in-patient care are being cared for in families and communities, but without the necessary financial support. Lack of such substantial resources at a family level has a negative impact, not only on the burden of care of the family caregiver but also on the burden of treatment as experienced by a mental health care user relative in this context. Burden of treatment (BTT) is a new model between sick people, their social networks and healthcare services. As burdens accumulate, some patients are overwhelmed, and the consequences are likely to be poor healthcare outcomes for the individual patients, increasing strain on family caregivers and rising demands and costs of health care services [10]. In the face of these challenges, we need to understand the resources that patients draw upon (including family caregiver support) as they respond to the demands of both burdens of illness and burdens of treatment and the ways that resources interact with healthcare utilization [10]. 

Although many families continue to provide care for their family members who have chronic mental illness, there is a dearth of studies on quantifying the burden of care for such family members, as well as the impact of such a burden on the carer’s physical, psychological and social health. This is especially true in rural communities, where caring for the sick is commonly left to the family members [11,12,13]. The purpose of this study was not only to quantify the burden of care among family caregivers who provide care for their relatives with chronic mental disorders, but also to explore possible associations between the caregiver burden of care and a range of caregiver and Mental Health Care User (MHCU) variables in the rural UMkhanyakude District in Kwa-Zulu Natal, South Africa. 

## 2. Research Methodology

### 2.1. The Study Design

The study used a quantitative survey to determine the burden of care among family caregivers of people who were diagnosed with psychotic (schizophrenia spectrum disorders) and mood (major depression and bipolar) disorders. 

### 2.2. Study Setting

The study was conducted in rural UMkhanyakude Health District, the second largest district in the Province of Kwa- Zulu Natal, with an estimated population of 625,846. The district has 53 primary health care facilities and a population of about 4400 mental health care users in the patient registers of these primary health care facilities. Because of this district’s rural setting, many people walk long distances to access basic health services, including mental health care services. The study was conducted in 30 health facilities that were identified by their various hospital managers as having a high number of mental health care users who collect medication for mental illness on a monthly basis.

### 2.3. Study Population

The study population consisted of primary caregivers of patients who live within the uMkhanyakude Health District and receive care from health facilities on an outpatient basis. 

### 2.4. Recruitment

The potential participants were recruited from identified health facilities. The potential participants consisted of individuals who were accompanying relatives with mental disorders that visited those health facilities for health reviews and to collect their medication. The inclusion criteria for the mental health care users were having received a diagnosis of schizophrenia, major depression, or bipolar mood disorder and attended treatment in the district facilities for at least a year. The inclusion criteria for the family members or study participants were that they be aged 18 years or older, serving as the primary caregiver for their mentally ill relative for at least a year, and able and willing to provide informed consent.

### 2.5. Sampling Techniques 

A purposeful and convenience sampling technique was used because all of the participants were linked to their mentally ill relatives who were accessing services.

### 2.6. Sample Size

Using the Raosoft sample size calculator for a population of 4400 mental health care users registered in the health facilities of the district, a 5% margin of error, a confidence level of 95%, and a response rate of 50%, a minimum sample size of 354 was calculated. The final sample size was 357. 

### 2.7. Data Collection Tools

(1)The Zarit Burden Interview (ZBI) Scale was used to measure the burden of care among the participants. The ZBI is a globally validated tool that is designed for measuring a caregiver’s perceived burden of care while providing family care for a relative. The tool has been widely used in both developed and developing countries and has been confirmed to be both reliable and valid [14,15,16].(2)A quantitative questionnaire was used to collect socio-demographic data of the participants, as well as data on their relatives with mental disorders.

### 2.8. Data Collection

Data were collected by the researcher and a research assistant, who was trained in the methodology of data collection, ethics and protocols to adhere to prevention of the spread of the SARS virus. Data collection took place over a period of 6 months (from October 2021 to April 2022). 

Data were collected in an interview room at each facility. The purpose of the study was explained to the group of potential participants in their own language, and they were given the opportunity to ask questions or seek clarifications. This was followed by the administration of the informed consent, which was followed by the administration of the socio-demographic questionnaire, and lastly the ZBI scale. All research documents were administered in IsiZulu, the local language understood by all participants. 

To accommodate the limited literacy and numeracy skills of many of the participants, matchsticks were used to demonstrate the concept of the Likert Scale for the ZBI. A table of five columns was drawn, with each column representing how the participant felt with regard to the item displayed by their mental health care user relative, during their home caring process. The first column did not have a match stick (representing never or none), one matchstick represented rarely, two match sticks represented sometimes, three matches represented quite a bit, and four match sticks represented extremely. 

The candidates were thanked and given a lunch pack as compensation for their time and participating in the study. 

### 2.9. Ethical Considerations

The proposal received ethical approval from both the Sefako Makgatho Health Sciences University Research Ethics Committee, (SMUREC/H/111/2021: PG), and the KZN Provincial DOH Research Committee (KZ_202109-022). Permission to conduct the study was obtained from the UMkhanyakude Health District Research Committee, the sub-District Hospital Management Executive Officers, and the operation managers of each participating health facility. The individual participants provided informed consent. 

## 3. Data Analysis

The raw data were captured into an Excel spreadsheet and uploaded into STATA version 14. The data of the caregivers and the patients were analyzed separately. Socio-demographic data were descriptively analyzed. The burden of care was determined by the scores on the ZBI scale, which has a maximum score of 88, with higher scores indicating heavier burden of care. The scores of the Zarit Burden Interview scale were used to categorize the total score of each participant as little or no burden (0–20), mild to moderate burden (21–40), moderate to severe burden (41–60) and extremely severe burden (61–88).The Pearson chi-square test was used to explore the association between a range of socio-demographic variables and burden of care as measured by the ZBI; (*p*-value = 0.05). Multivariate logistic regression was used to explore the association between the socio-demographic variables that were significantly associated with burden of care, based on the chi-square test. 

## 4. Characteristics of Caregivers

The mean age of the participants was 50.3 years, with the youngest being 18 years of age and the oldest caregiver 65. The greatest proportions of the sample were female (n = 306, 85.71%), single (*n* = 192, 53.78%) and unemployed (*n* = 301, 84.31%). Almost all of the participants had daily contact (*n* = 356, 99.72%) with the patient, but only 15.97% (*n* = 57) reported receiving help with their care-giving duties. The average household monthly income of the participants was ZAR_3803.70. Further details are provided in Table 1 below.

### 4.1. Socio-Demographic Information of MHCUs

The majority of patients were between the ages of 26–40 years (*n* = 157, 43.98%), male (*n* = 245, 68.63%) and suffering from schizophrenia (*n* = 213, 59.66%). Nearly all of the patients were unemployed (*n* = 356, 99.72%), with 77.31% (*n* = 276) of them receiving a disability grant. Table 2 below provides further details on the socio-demographic variables.

### 4.2. Quantification of Caregiver Burden

The results showed that 89.64% (*n*=303) of the caregivers were experiencing caregiver burden, with a mean ZBI score of 41.60 when a cut-off point of <21 was utilized. A majority of the participants were experiencing mild/moderate levels of burden (*n* = 141, 39.50%), followed by 35.01% (*n* = 125) that reported moderate/severe levels and 15.13% (*n* = 54) that experienced severe levels of caregiver burden. Figure 1 below shows the prevalence of the caregiver burden and further illustrates the findings. 

The heaviness of the burden of care ranged from little to severe, as shown in Figure 2 below.

### 4.3. Factors Associated with Care giving Burden

The Pearson chi-square test of association showed that there were seven main factors that were associated with the care-giving burden. Three of those factors were directly related to demographic variables of the caregiver, i.e., age, help received with care giving role and self-reported depression (*p*-value = 0.05). Three other factors were related to the employment status, gender, and relapsed admission history of the mental health patient (*p* = 0.05). The remaining factor was related to the monthly household income (*p*-value = 0.05). Table 3 below illustrates factors associated with care-giving burden. 

On multivariate logistic regression, only monthly family income and relapsed after admission remained statistically significant, as shown on Table 4 below: 

## 5. Discussion

The finding that most family caregivers were female was previously reported in other studies conducted both in developing and developed countries, where care-giving responsibilities were assumed mostly by females [5,17,18], and the burden of care was higher among females [19,20]. Generally, the burden of care is particularly high among middle-aged females who find themselves obligated to care for their children and their aging parents [20,21]. The Danish study, in trying to understand how smart objects (e.g., assistive robots and various alarm systems) could actively reshape the everyday practices in families with elderly consumers, agrees with the finding that care-giving is widely viewed as an inherently female responsibility [21]. Females in general are exposed to norms and societal expectations of being good women when growing up, despite the availability of ambient assistive living technologies in developed world settings. They not only strive to be good parents to their sick children, but also to be good children to their sick parents, suffering from either physical or psychiatric conditions [21] This, therefore, suggests that female caregivers in various socio-economic contexts need additional resources to support their mental health, such as social support as well as psycho-educational support.

The current study found that males were more affected by psychotic and mood disorders, aligning with other studies that reported that major depression and schizophrenia spectrum disorders were more prevalent among males [22], and they were the ones mostly looked after by females. These findings concur with some sub-Saharan studies performed on the burden of care in general and mental health care specifically, that reported that males were mostly affected by serious mental illness, especially in the psychotic spectrum range [23,24]. This suggests that a focus on screening for these disorders among men should be strategically integrated into men’s health services.

Of interest to this research was the “cohabiting group” of family caregivers (*n* = 76 − 21%), whose roles are supposed to be the same as those of the married group because they live with their partners permanently, although not officially married, which suggests limited commitment. This group formed a quarter of the sample, and its marital status, under trying circumstances, can jeopardize the quality of care and dedication given to the supposed mental health care user spouse or partner. This cohabiting concept had no literature support and its subsequent impact on the home care of mental health care users, Noticeably, a significant majority (88%, *n* = 313) of the mental health care users were found to be single, which is similar to the findings reported by other studies, i.e., most people with mental disorders are single [25]. This can be explained by the difficult social situations experienced by people with mental disorders, which render them unable to form and maintain social relationships, as well as the stigma perpetrated against mental illness and people so affected [26]. In the current study, marital status for either caregivers or their mental health care user relatives was perceived by caregivers as not contributing significantly to their burden of care.

The majority (83%, *n* = 301) of the family caregivers in the current study were unemployed, which confirms the high unemployment rates in South Africa [27,28], especially among Black Africans [29] who live in rural areas [30]. Some family caregivers had to quit their jobs in order to fully take care of their mentally ill relatives. This concurs with the findings of studies from both developed and developing countries, which reported that family caregivers were often compelled by circumstances and demands to care for their mentally ill relatives, to the extent that they often had to quit their jobs in order to offer full time care, despite the poverty this decision could expose them to [31,32,33]. Moreover, it helps to improve the morale of the family caregiver if the mental health care user is employed because relatives believe that if the mentally ill person has a job, it suggests that he/she is being cured of his/her mental condition, as well as improving his/her dignity and self-esteem [1]. In this study, most of the unemployed mental health care users (77%, *n* = 276) depended on Government disability grants. These social grants contribute significantly toward the family monthly income, thus improving the socio-economic status of the family [5].

Only a few participants (4%, *n* = 14) reported that they were depressed because of caring for their mental health care user relatives, which proved to be statistically significant and, therefore, contributed a great deal towards their burden of care. This finding concurs with an Asian study that reported that depression could affect caregivers of mental health care user relatives in two ways, i.e., either by easing their burden of care or increasing depressive symptoms they already exhibited during the caring process [34].

The results of the current study showed that almost half of the sample had been primary caregivers for long periods of between 11 and 30 years, during which time they were living with the family member being cared for, almost every day. This situation statistically proved not to be significant as far as family member caregiver burden was concerned. The literature acknowledges the positive impact of family support on the quality of care given to the mental health care user relative, within the family context [35,36].

Within the context of the family dynamics of the Zulu culture, the relationship between the caregiver and the MHCU does not mean much because it is not only the close family member but also the extended family member assumes the burden of caring if the worst comes to the worst. These family dynamics differ from those in developed countries, where the nuclear family does not necessarily embrace the extended family members [20,37].

The sample had fewer MHCUs who were diagnosed with major depression, which contrasts with global trends. This may be explained by previous findings that in comparison with other countries, South Africa has lower rates of major depression [38]. But, it may also be explained by under-diagnosis and under-reporting of depression, which was reported to be up to 87% in developing countries [39]. Either way, the need for financial and human resources for diagnosing, treating and managing depression remains high, with the ratio of psychiatrists to a given unit of population being unfavorable [40,41]. Moreover, major depression is not readily diagnosed because it is mostly limited to the experience of the patient, whereas the destruction and dramatic acts which are displayed by patients with schizophrenia or bipolar mood disorder (with manic episodes) demand more attention from society and apparently add to the burden of the caregiver [42]. Although most of the MHCUs had not experienced recent relapses, relapse had more negative impact on the mental health of the caregiver, and was, therefore, statistically associated.

## 6. Study Limitations

The mental health care users themselves were not allowed to be present during the interviews in order to guarantee the confidentiality of the information. This study was limited to family members whose relatives were diagnosed with psychotic disorders (schizophrenia spectrum disorders) and mood disorders (major depression and bipolar mood disorders), because these categories have enduring residual symptoms that impact the quality of care being rendered. Due to the severity of their symptoms and the decline in their functional levels, they need continuous care. This continuity of care in the midst of scarce resources brings about the increased burden experienced by family members. Such burden expresses itself in disease-related factors, clinical-related factors, socio-demographic as well as psycho-social factors. Better outcomes for this group of disorders are related to the presence of social and financial support and less severity of mental illness.

## 7. Conclusions

Although the difficulties experienced by family caregivers of MHCUs have been reported [43], this study specifically quantified the burden of care, which integrated not only the mental aspects, but a comprehensive burden that included other aspects like finances, access to additional assistive resources, and other responsibilities that are on the shoulders of the caregivers. Both the prevalence and severity of mental disorders are reported to be increasing globally; however, in South Africa, the resources allocated to the treatment of mental disorders have not increased proportionally [43]. There is a high need for financial and human resources, particularly in developing countries (South Africa included) for diagnosing, treating and managing depression within societies. The study findings, therefore, acknowledge the mental distress of family caregivers and concur with other studies that MHCUs need additional resources to adequately attend to their needs and thus reduce the burden on family caregivers.

## 8. Recommendations

Given the high burden of care faced by family caregivers, it is recommended that mental health professionals at the primary health care level align their service delivery plans with the identification of the needs of family caregivers and refer these to social services that can benefit the MHCU. It is further recommended that screening for mental disorders, especially in the psychotic spectrum range, should be integrated into health services at the primary health care level. Without attending to this important aspect of community-based care, the treatment outcomes for MHCUs will remain poor. Acknowledging the extensive mental health impacts of the COVID-19 pandemic on society, there is a need to incorporate care for those mental health impacts into the broader health care services.

## Figures and Tables

**Figure 1 healthcare-11-02686-f001:**
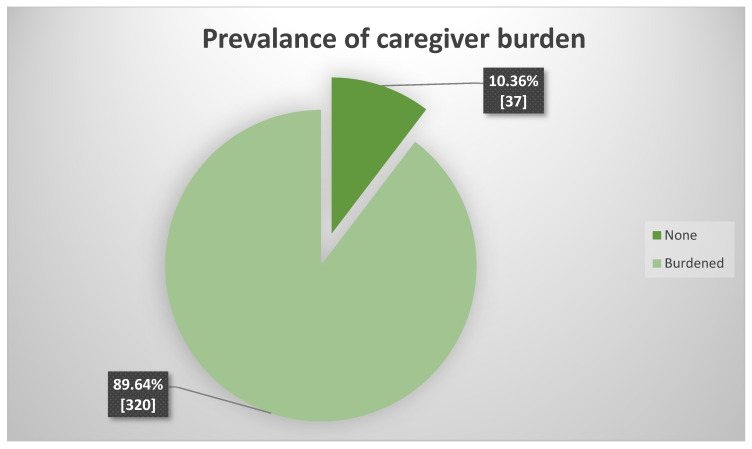
Prevalence of caregiver burden.

**Figure 2 healthcare-11-02686-f002:**
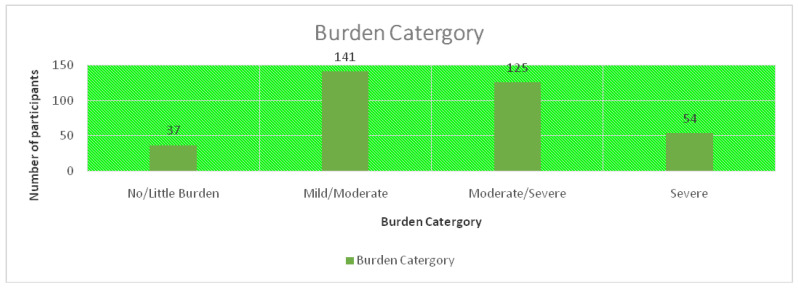
The levels of burden of care.

**Table 1 healthcare-11-02686-t001:** Characteristics of caregivers.

Variable	Frequency(*n*)	Percentage (%)
**Age (*n* = 357)**		
≤40 years	77	21.57
41–60 years	185	51.82
≥61 years	95	26.61
**Age (mean 50.3; SD 12.2; min 18; max 65)**		
**Gender (*n* = 357)**		
Female	306	85.71
Male	51	14.29
**Level of education (*n* = 357)**		
No formal education.	99	27.73
Primary	110	30.81
Secondary	127	35.57
Tertiary	21	5.88
**Marital status (*n* = 357)**		
Co-habiting	76	21.29
Married	80	22.41
Single	192	53.78
Widowed	9	2.52
**Employment status (*n* = 357)**		
Employed	56	15.69
Unemployed	301	84.31
**Religion (*n* = 357)**		
Christian	323	90.48
Nazareth	25	7.00
None	3	0.84
Other	6	1.68
**Other chronic diseases (*n* = 357)**		
No	153	42.86
Yes	204	57.14
**Self-reported depression (*n* = 357)**		
No	343	96.08
Yes	14	3.92
**Number of children**		
None	14	3.92
1–4 children	264	73.95
5 children or more	79	22.13
**Monthly family income**		
Below 2000	146	40.90
ZAR_ 2001–4000	147	41.18
ZAR_4000–10,000	48	13.45
Above ZAR_ 10,000	16	4.48
**Income (Mean,ZAR_ 3803.70; SD 4217.45; Min,ZAR_0; Max ZAR_39,000 )**		
**Relationship to patient**		
Child	135	37.82
Parent	23	6.44
Sibling	85	23.81
Spouse	39	10.92
Other	75	21.01
**Receiving help with care-giving (*n* = 357)**		
No	300	84.03
Yes	57	15.97
**Living with patient (*n* = 357)**		
No	16	4.48
Yes	341	95.52
**Other family members needing help (*n* = 357)**		
No	216	60.50
Yes	141	39.50
**Frequency of contact with patient (*n* = 357)**		
Everyday	356	99.72
Occasional	1	0.28
**Number of years as a caregiver**		
Less than 5 years	83	23.25
6–10 years	91	25.49
More than 10 years	183	51.26
**Care-giving years (Mean 14; SD 9.04; Min 1; Max 54)**		

**Table 2 healthcare-11-02686-t002:** Socio-demographic information of MHCUs.

Variable	Frequency	Percentage (%)
**Age (*n* = 357)**		
Below 25 years	39	10.92
26–40 years	157	43.98
41–60 years	131	36.69
Above 60 years	30	8.40
**Age (mean 40.5; SD 12.6; min 19; max 76)**		
**Gender (*n* = 357)**		
Female	112	31.37
Male	245	68.63
**Level of education (*n* = 357)**		
No Education	62	17.37
Primary	130	36.41
Secondary	152	42.58
Tertiary	13	3.64
**Marital status (*n* = 357)**		
Co-habiting	28	7.84
Married	16	4.48
Single	313	87.68
**Diagnosis (*n* = 357)**		
Bipolar mood disorder	103	28.85
Major depressive disorder	41	11.48
Schizophrenia	213	59.66
**Duration of illness (*n* = 357)**		
Less than 5 years	74	20.73
5–10 years	93	26.05
11–20 years	126	35.29
More than 20 years	64	17.93
**Relapsed admission (*n* = 357)**		
No	299	83.75
Yes	58	16.25
**Disability grant**		
No	81	22.69
Yes	276	77.31
**Employment status (*n* = 357)**		
Employed	1	0.28
Unemployed	356	99.72
**Relationship to caregiver (*n* = 357)**		
Child	24	6.72
Other	75	21.01
Parent	135	37.82
Sibling	86	24.09
Spouse	37	10.36

**Table 3 healthcare-11-02686-t003:** Factors associated with care-giving burden.

Factors	Frequency (%)	Burdened (%)	Not Burdened (%)	Chi²	*p*-Value
**Age of Caregiver**				**10.1653**	**0.006**
≤40 years	77 (21.57)	65 (20.31)	12 (32.43)		
41–60 years	185 (51.82)	175 (54.69)	10 (27.03)		
≥61 years	95 (26.61)	80 (25.00)	15 (40.54)		
**Monthly family income**				**20.6410**	**0.000**
Below 2000	146 (40.90)	139 (43.44)	7 (18.92)		
ZAR_2001–4000	147 (41.18)	133 (41.56)	14 (37.84)		
ZAR_4000–10,000	48 (13.45)	35 (10.94)	13 (35.14)		
Above ZAR_10,000	16 (4.48)	13 (4.06)	3 (8.11)		
**Self-reported depression of caregiver**				**5.1997**	**0.023**
No	343 (96.08)	310 (96.88)	10 (3.13)		
Yes	14 (3.92)	33 (89.19)	4 (10.81)		
**Receiving help with care-giving role**				**5.8278**	**0.016**
No	300 (84.03)	274 (85.63)	46 (14.37)		
Yes	57 (15.97)	26 (70.27)	11 (29.73)		
**Gender of the patient**				**4.0719**	**0.044**
Female	112 (31.37)	95 (29.69)	17 (45.95)		
Male	245 (68.63)	225 (70.31)	20 (54.05)		
**History of relapse after admission**				**5.5647**	**0.018**
No	299 (83.75)	263 (82.19)	36 (97.30)		
Yes	58 (16.25)	57 (17.81)	1 (2.70)		
**Employment status of MHCU**				**8.6729**	**0.003**
Employed	1 (0.28)	0 (0.00)	1 (2.70)		
Unemployed	356 (99.72)	320 (100.00)	36 (97.30)		

**Table 4 healthcare-11-02686-t004:** Multivariate analysis.

Factors	Coef.	Std. Err.	*p*>|z|	[95% Conf. Interval]
Age of caregiver	0.0204188	0.2593118	0.937	−0.487823 0.5286606
Monthly family income	−0.7151498	0.2243773	0.001	−1.154921 −0.2753784
Self-reported depression of caregiver	−1.225815	0.6799649	0.071	−2.558521 0.106892
Receiving help with care-giving role	−0.2534951	0.4568942	0.579	−1.148991 0.6420011
Gender of the patient	0.5283181	0.3777569	0.162	−0.2120719 1.268708
Relapsed admission patient history	2.248435	1.056517	0.033	0.1776989 4.31917

## Data Availability

The data presented in this study are available on request from Sefako Makgatho Health Sciences University. The data are not publicly available because the University does not have a platform to avail its data to the public, yet.

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
