# Peer review of "Burden of Care of Family Caregivers for People Diagnosed with Serious Mental Disorders in a Rural Health District in Kwa-Zulu-Natal, South Africa"

_healthcare, 2023, doi:10.3390/healthcare11192686_

Round 1

Reviewer 1 Report

Thank you for inviting me to review this manuscript entitled"Burden of care of family caregivers for people diagnosed with 2 serious mental disorders in a rural Health District in Kwa- 3 Zulu-Natal, South Africa" which aimed to quantify the burden of care among family caregivers of relatives with serious mental 14 disorders, as well as to explore possible association between the caregiver burden of care and a 15 range of caregiver and Mental Health Care User (MHCU) variables in a rural district in Kwa-Zulu 16 Natal, South Africa. T

It is a very well-written paper with a good flow. 

Minor comments

Please add more details about the survey used?

Administration language?

Data recruitment period? 

Results

did you run the Univariate analysis before the Multivariate analysis? please add the results as supplementary material 

discussion

Please add the clinical implications and study limitations at the end of this section 

Author Response

Thank you so much for initially reviewing our article and also commenting on the previously submitted point by point corrections to the comments.

Clinical Implications and Study limitation additions have been highlighted in the coverletter and also appear in the main document (highlighted, with lines allocated). In fact, all the corrected comments are highlighted in the main document.

Reviewer 2 Report

This manuscript investigates the burden of care of family caregivers in the context of mental disorders. This is an interesting, timely, and relevant topic. The manuscript, however, requires some improvements before it can be considered publishable.

In the abstract, in line 20, "tested positive for" sounds like a clinical assessment. Maybe rephrase it as "found to be affected by" or something similar?

Please reformulate the sentence in lines 33 to 36. It is too long. Consider splitting it into two sentences.

In line 38, it is unclear whether “caregivers” refers to formal ones or family ones. In general, when applying the concept of “caregivers” throughout the entire manuscript, please specify the type of caregivers described or discussed.

In line 47, you change from “burden of disease” to “burden of care”. If not necessary, please do not use two different terms. The concept of burden of care should also be defined more clearly – right now this is left to the reader. It would also help to relate and delineate it from the concept of burden of treatment as used in [NEW2].

[NEW2] Rethinking the patient. https://doi.org/10.1186/1472-6963-14-281

In line 76, while describing the purpose of the study, you should be more explicit. Just quantifying the burden of care is not precise enough and does not do justice to your work.

The paragraph in lines 203-205 makes an important observation that should be strengthened by adding both another geographic and care context (elderly in Europe) to the existing ones (cancer patients in North America; children with AIDS in Africa; mental disorders in South America). A possible formulation and reference could be:

The finding that most family caregivers were female was previously reported in other studies, where caregiving responsibilities were assumed mostly by females [14, 5]. The burden of care is particularly high among middle-aged females who find themselves obligated to care for their children [22] and their ageing parents [NEW1].

[NEW1] Reassembling the elderly consumption ensemble. https://doi.org/10.1080/0267257X.2022.2078862

Please, fix reference 1 and follow the style used for the other references.

Mostly ok. Conceptual rigour might be improved, though. See the comments above.

Author Response

Thank you again for your time and patience in reviewing our article for the second round. We attach a Word Document with suggested comments and corrected responses. All the corrections are in red highlight in the manuscript. 

Round 2

Reviewer 2 Report

The authors have addressed the concerns regarding the content of the manuscript.

There are some minor language and editing issues that need to be taken care of. As an example, the new references have not been formatted in the same styles as the previous ones. May et al. needs to be extended to list the authors. The second reference uses full given names in front of the surnames. The given names should be abbreviated and put after the surnames.
In Line 117, it sounds as "These" refers to the "facilities" rather than the "participants". Please reformulate.
Also, Line 232, it should be "agrees with" instead of "agrees to".

Author Response

Thanks for reviewing our article for the second round. A point by point response on reviewer's comments have been attached. Corrected comments are highlighted in red in the manuscript.
